# Pathways for Sensing and Responding to Hydrogen Peroxide at the Endoplasmic Reticulum

**DOI:** 10.3390/cells9102314

**Published:** 2020-10-18

**Authors:** Jennifer M. Roscoe, Carolyn S. Sevier

**Affiliations:** Department of Molecular Medicine, Cornell University, Ithaca, NY 14853, USA; jr953@cornell.edu

**Keywords:** endoplasmic reticulum (ER), hydrogen peroxide, reactive oxygen species (ROS), redox signaling, cysteine oxidation, BiP, IRE1, SERCA2, unfolded protein response (UPR)

## Abstract

The endoplasmic reticulum (ER) has emerged as a source of hydrogen peroxide (H_2_O_2_) and a hub for peroxide-based signaling events. Here we outline cellular sources of ER-localized peroxide, including sources within and near the ER. Focusing on three ER-localized proteins—the molecular chaperone BiP, the transmembrane stress-sensor IRE1, and the calcium pump SERCA2—we discuss how post-translational modification of protein cysteines by H_2_O_2_ can alter ER activities. We review how changed activities for these three proteins upon oxidation can modulate signaling events, and also how cysteine oxidation can serve to limit the cellular damage that is most often associated with elevated peroxide levels.

## 1. Introduction

All cells are susceptible to oxidative damage. Damage often appears concomitant with a buildup of reactive oxidants and/or a loss of antioxidant systems. In particular, an accumulation of cellular reactive oxygen species (ROS) has attracted much attention as a source of cellular damage and a cause for a loss of cellular function [1]. In keeping with these observations, most historical discussions of ROS focus on the need to defend against the toxic and unavoidable consequences of cellular ROS production, in order to limit cellular dysfunction and disease. In particular, substantial attention has been paid to the potential accumulation of ROS as a byproduct of cellular respiration, and the importance of detoxification pathways that limit mitochondrial ROS accumulation and ensuing damage.

Yet, in the last two decades, the view of ROS as a toxic mitochondrial derivative has evolved. An appreciation of the diversity of ROS, the benefits provided by the action of some ROS as signal molecules, and the variety of cellular ROS sources (beyond the mitochondria) have begun to permeate the literature (e.g., see [2,3,4]). These new views regarding ROS have begun to refocus the discussion of ROS production and utilization by cells.

Here we concentrate our attention on the current view of hydrogen peroxide (H_2_O_2_), one type of ROS, and the endoplasmic reticulum (ER), an emerging source of H_2_O_2_ production and H_2_O_2_-based signaling events. We begin with a discussion of the properties of H_2_O_2_ and sources of ER-localized peroxide. Later, we highlight examples of ER-based signaling events involving H_2_O_2_. We focus on three targets of reversible modification by ER peroxide: the molecular chaperone BiP, the transmembrane stress-sensor IRE1, and the calcium pump SERCA2. We discuss how reversible post-translational oxidation of cysteine residues in BiP, IRE1, or SERCA2 by peroxide alters protein function, and how altered activities for these proteins can help the ER adapt to rising peroxide levels.

## 2. Properties of H_2_O_2_

Successive reduction of molecular oxygen (O_2_) generates multiple distinct oxygen-containing species, including H_2_O_2_, the superoxide anion (O_2_^•–^), and the hydroxyl radical (^•^OH) (Figure 1). Collectively, these molecules are referred to as ROS, emphasizing the chemical reactivity of these oxygen-containing molecules. The hydroxyl radical (known also as a “free radical”) is considered more reactive, less stable, and more destructive to macromolecules than H_2_O_2_ [4,5]. Conversely, H_2_O_2_ is considered a strong two-electron oxidant, but poorly reactive with most macromolecules [3,6].

The limited, slow reactivity of H_2_O_2_ with most biological molecules does not mean that cells are unaffected by elevated levels of intracellular H_2_O_2_. H_2_O_2_ is easily converted to the highly reactive hydroxyl radical via a Fenton or a Fenton-like reaction catalyzed by a transition metal [6,7,8], and it is widely considered that the cellular damage associated with peroxide exposure is attributed to the formation of hydroxyl radicals mediated by intracellular metals. The presence of cellular H_2_O_2_ can also be coincident with superoxide-induced damage; the spontaneous or catalyzed dismutation of superoxide generates H_2_O_2_, connecting the presence of these species [9].

The physiochemical properties of H_2_O_2_ are similar to those of water. Aquaporin (AQP) channels, first established to mediate the diffusion of water across membranes [10,11], have been shown to facilitate transmembrane H_2_O_2_ movement [12]. The mammalian AQP11 has been localized to the ER and established to serve as a “peroxiporin”, facilitating the movement of H_2_O_2_ across the ER membrane [13,14]. A role for mammalian AQP8 in the movement of H_2_O_2_ at the ER has been shown as well [15]; however, localization data suggest that the ER activity of AQP8 reflects a subset of the AQP8 found in the ER in transit to the plasma membrane [13]. Many aquaporins are regulated at the post-translational level in response to cellular or environmental changes [16,17]. Post-translational modulation of the ER-localized AQP11 remains to be established, but data for other peroxiporins (including AQP8 [18]) suggest the intriguing possibility for regulated transport of H_2_O_2_ across the ER membrane.

Despite the generally poor reactivity of H_2_O_2_ with most macromolecules, peroxide does show reactivity towards select protein thiols. Central to most characterized redox sensing, signaling, and regulation events is the direct oxidation of specific protein cysteine residues by H_2_O_2_, generating a sulfenic acid adduct (–SOH) (Figure 2). The sulfenic acid can transition further to other oxidized cysteine forms, including glutathionylated (–SSG) and disulfide-bonded species. Further oxidation of sulfenic acid by peroxide can also yield sulfinylated (–SO_2_H) and sulfonylated (–SO_3_H) proteins. A means to reduce sulfonylated proteins in cells has not been identified. The only established route for sulfinic acid reduction is through the action of the enzyme sulfiredoxin (SRX) [19]. SRX is considered key to maintain the catalytic activity of peroxiredoxins [20], and additional new targets of SRX are emerging [21]. Interplay between H_2_O_2_ and other signaling molecules, including hydrogen sulfide (H_2_S) and nitric oxide (NO), can generate further oxidants that can facilitate cysteine oxidation (e.g., peroxinitrite) and additional cysteine modifications (e.g., persulfides) and signaling outcomes [22]. The conversion of sulfenic acid to reversible forms (glutathione adducts, disulfides, persulfides) is thought to safeguard against cysteine overoxidation by H_2_O_2_, and the formation of the (generally) irreversible sulfinic and sulfonic acid (e.g., [23]).

The reversibility of cysteine oxidation allows for transient metabolic changes and/or signaling events, and distinguishes cysteine oxidation (signaling) from the irreversible oxidation of amino acid side chains and protein backbone carbonyls (damage). Much like other post-translation modifications, reversible cysteine oxidation can change protein function by altering enzyme activities and/or changing macromolecular interaction (for further review see [24,25,26,27]). Sites of reversible cysteine oxidation are also colloquially referred to as redox switches.

The movement of H_2_O_2_ across membranes, alongside its relative stability and poor reactivity, enables peroxide generated in one area of the cell to impact a different compartment; these properties empower H_2_O_2_ as a signaling molecule. How far H_2_O_2_ can move and signal from the initial generation site will be limited by the presence of antioxidants and/or enzymes that mediate peroxide breakdown (peroxidases) along the path of diffusion. In general, H_2_O_2_ exhibits relatively fast reaction with peroxidase active-site cysteines [28,29]. Thus, the ability of H_2_O_2_ to react with non-peroxidase thiols requires either a target reaction that exceeds the rate of reaction with enzymes that mediate peroxide breakdown and/or movement of peroxide within regions with less robust antioxidant defenses. As our view of H_2_O_2_ has evolved to include H_2_O_2_ as more than a damaging oxidant, so has our view of the consumers of H_2_O_2_. Peroxidases such as peroxiredoxin and catalase should be considered not only safeguards protecting cells from reaching a state of oxidative stress by facilitating the removal of peroxides but also as agents that serve to control the availability of peroxide for signaling.

Robust quantification of basal cellular H_2_O_2_ levels and the fold-changes associated with various physiological and environmental conditions remain a work-in-progress for the redox field. Many challenges are associated with the available small-molecule and protein-based H_2_O_2_ probes [22,30,31,32,33,34]. In addition, technical challenges are associated with the presence of H_2_O_2_ and/or H_2_O_2_-producing molecules in common reagents, including growth media and detergents [35,36]. Although the reported measurements of cellular H_2_O_2_ vary, a general consensus has emerged that steady-state intracellular H_2_O_2_ is maintained in the sub-micromolar (nM) range [37,38,39,40] whereas cellular toxicity is associated with micromolar concentrations at the whole cell level [37]. Although these measurements come with limitations, they provide support for the general prevailing idea that low (physiological) levels of H_2_O_2_ may be of benefit for signaling within cells, whereas high (pathological) levels may exert damage and result in cell death.

## 3. Sources of ER-localized H_2_O_2_

Enzymes that generate H_2_O_2_ exist within the ER lumen. The diffusibility of H_2_O_2_ means also that ROS generated throughout the cell, as well as extracellular ROS exposure, have the potential to impact molecules at the ER. As discussed above, the high reactivity of H_2_O_2_ with enzymes in the peroxidase family (and the turnover of H_2_O_2_ by these enzymes) implies the distance traveled by H_2_O_2_ from its site of origin may be limited by the presence of detoxification enzymes along the diffusion path. Thus, H_2_O_2_ sources most anticipated to directly impact the ER are likely close in physical space. Below we outline some of the established sources for ER H_2_O_2_ as well as some proximal sources that we speculate could also directly influence the ER redox environment (Figure 3).

### 3.1. ER-Localized Enzymes

The most readily appreciated sources of ER H_2_O_2_ are oxygen-utilizing enzymes that localize to the membrane or lumen of the ER. Arguably, the most recognized enzymatic source of ER peroxide is the flavoprotein ERO1. Together with members of the PDI family of oxidoreductases [41], ERO1 facilitates the formation of disulfide bonds in folding nascent polypeptides [42]. Electrons removed during the formation of a disulfide bond are transferred from the nascent secretory protein to ERO1 via PDI. H_2_O_2_ is produced when ERO1 transfers the electrons received from a PDI to molecular oxygen [43,44]. In mammals, H_2_O_2_ generated at the ER is turned over by the enzymes peroxiredoxin IV and the glutathione peroxidases GPx7/8, and the activity of these enzymes is productively coupled to PDI oxidation and serves to augment disulfide bond formation [45,46,47,48].

If one presumes that every disulfide bond yields a molecule of peroxide, back-of-the-envelope calculations suggest that copious amounts of H_2_O_2_ will be generated as a byproduct of oxidative folding [49,50]. Measurements suggest a relatively high (~0.7 µM) basal H_2_O_2_ concentration in the ER of HeLa cells [51]. The enhanced oxidation of a peroxide-responsive biosensor (HyPer) targeted to the ER (relative to HyPer targeted to other organelles) further supports a relatively high level of ER H_2_O_2_ [52]. Yet, the quantification of H_2_O_2_ by the ER-localized HyPer probe must be considered carefully given the challenges associated with disulfide-based sensors targeted to the ER lumen, where PDI may facilitate oxidation of the sensor disulfide, which can be misinterpreted as H_2_O_2_-mediated oxidation of the probe [52,53]. Progress towards overcoming the limitations of the original ER-localized HyPer have been made; introduction of an additional cysteine into the original HyPer probe (TriPer) has been shown to attenuate the responsiveness of HyPer to PDI [54].

NADPH oxidases (NOX) and dual oxidases (DUOX) family members are also sources for ER H_2_O_2_. The majority of the NOX/DUOX family members are multi-subunit transmembrane enzyme complexes, which include activating and regulatory domains that come together to generate ROS to facilitate redox-signaling events (more extensively reviewed in [55]). Most NOX isoforms are reported to couple NADPH oxidation to the direct production of superoxide, which can dismutate to H_2_O_2._ An exception is mammalian NOX4, which is suggested to have the capacity to generate H_2_O_2_ directly [56]. Multiple NOX proteins have been localized to the ER, but it remains unclear whether the detection of these NOX at the ER reflects a transient or steady-state ER localization. At present, the isoforms most accepted as predominantly ER-localized NOX are mammalian NOX4 [57] and the yeast Yno1/Aim14 [58], although the subcellular localization of NOX4 may not be limited to the ER [59,60]. As the ER appears to be an activation compartment for several isoforms, NOX at the ER in transit to the surface may also contribute to ER ROS production.

Experimental outcomes for NOX4 [61], and topology predictions for Yno1/Aim14 [58], suggest that the C-terminal globular domain is oriented into the cytoplasm for both these ER-localized NOX, releasing superoxide and/or H_2_O_2_ into the cytoplasm. In general, NOX are a well-established mechanism for the generation of high local ROS concentrations, and these enzymes are anticipated to contribute towards substantial levels of proximal ER ROS. The oxidoreductase PDI has been shown to influence the activities of several NOX isoforms [62], demonstrating an additional case where the PDI oxidoreductases are coupled to ROS generation and/or turnover.

Breakdown of endogenous and exogenous substrates by the diverse members of the cytochrome P450 (CYP) family of heme monoxygenases also generates superoxide and H_2_O_2_ [63,64]. CYP is a large protein family of soluble and transmembrane proteins. Most characterized CYP are located at the ER membrane (microsomal CYP), although CYP members are found also at the mitochondria and plasma membrane. CYP are most recognized for their role in the metabolism of exogenous xenobiotics and drugs; CYP also mediate important endogenous activities, including the biosynthesis of steroids, sterols, and fatty acids [65]. Early studies established the NADPH-dependent production of H_2_O_2_ by liver microsomal CYP [66]. It is now appreciated that various steps in the CYP reaction cycle can generate H_2_O_2_ and superoxide [63,64]. The CYP active site appears oriented towards the cytoplasm [67], although the general CYP topology is still debated.

### 3.2. Mitochondria-Associated Membranes

Mitochondria are arguably the most recognized intracellular sources of ROS. Healthy mitochondria are a major source of superoxide, which can be produced stochastically through the inefficient transfer of electrons at several points in the electron transport chain (ETC) [68]. Superoxide generated in the mitochondria can be dismutated to H_2_O_2_ by the mitochondrial superoxide dismutases located in the mitochondrial matrix (SOD2) and intermembrane space (SOD1). The enzymes that facilitate disulfide bond formation in the inner membrane space also generate H_2_O_2_ as a byproduct [69,70]. In an analogous system to ERO1-PDI pathways, electrons removed during the formation of a disulfide bond in a mitochondrial protein are transferred to the flavoprotein Erv1/ALR via the oxidoreductase Mia40/CHCHD4. Erv1/ALR can generate H_2_O_2_ upon transfer of electrons to molecular oxygen, although a buildup of H_2_O_2_ as a consequence of oxidative folding is limited in part by use of cytochromes as an alternative electron acceptor [71,72,73,74]. In addition, the mitochondria contain several peroxidases that convert H_2_O_2_ to water [75].

Multiple sites along the vast ER membrane form tight physical contacts with the mitochondria; these dynamic linkages are termed mitochondria-associated membranes (MAMs). MAM are recognized for their role in Ca^2+^ and lipid exchange between the ER and mitochondria [76]. Several sources and modulators of ROS have been localized to the MAM, including ERO1 [77]. As the mitochondria and ER are both sites of ROS production and turnover, it seems logical that MAMs would also serve as a site of H_2_O_2_ exchange. Nanodomains of H_2_O_2_ at MAMs have been detected upon the efflux of Ca^2+^ from the ER [78]. Peroxide accumulation was shown to trigger further ROS mobilization at the mitochondria (likely via the ETC), which was communicated to the ER to facilitate additional ER Ca^2+^ efflux [78]. Although direct action of the mitochondrial peroxide at the ER membrane was not established, it was speculated that the release of ER Ca^2+^ stores in response to mitochondria H_2_O_2_ reflects the direct modification and modulation of specific thiols in the ER IP_3_ receptor by H_2_O_2_ [78].

In general, the specifics as to when mitochondrial H_2_O_2_ can diffuse and impact surrounding organelles, including the ER, has not been fully resolved. Notably, a recent study using a newer, more sensitive H_2_O_2_ sensor (termed HyPer7) concluded that H_2_O_2_ does not readily diffuse out of the mitochondria matrix into the cytoplasm, suggesting that the majority of H_2_O_2_ generated in the matrix will not reach and influence the ER [79]. These studies followed ROS generated by the chemogenetic substrate-controlled D-amino oxidase (DAO), which arguably models a level of physiological ROS. Alternatively, release of a burst of ROS from the mitochondria (likely H_2_O_2_) has been established to occur through a process termed ROS induced ROS release (RIRR) [80]. The biphasic RIRR process includes the slow generation of trigger ROS, which leads to a subsequent ROS burst upon the dissipation of the mitochondrial membrane potential. The regulated opening of mitochondrial pores on the outer membrane allows for ROS release from the mitochondria, which can diffuse to other proximal organelles, such as the ER and neighboring mitochondria. The release of ROS from a mitochondrium, and the detection of this ROS by surrounding mitochondria, has been highlighted as a means of signal amplification, converting a lower amount of ROS into a pathological ROS, through the further stimulation of superoxide production by the ETC. Overall, a role for the triggered release of mitochondrial ROS release under select conditions to initiate signaling beyond the mitochondria is emerging, with a potentially lesser role for released basal ETC ROS in signaling outside of the mitochondria.

### 3.3. Additional Proximal Sources of ER H_2_O_2_

The vast ER membrane network makes physical contact with numerous additional organelles beyond the mitochondria. Regions of close contact between organelles have been generally termed membrane contact sites (MCS), and (as true for the MAMs) these stable or transient sites of physical contact with the ER are thought to enable the efficient transfer of metabolites between compartments [81]. Significant enzymatic sources for H_2_O_2_ exist distributed throughout the cell, including plasma membrane NOX [82] and peroxisomal beta-oxidation [83], and intriguing connections have been made between these local sites of ROS generation and ER function. For example, activity of the plasma membrane-localized NOX2 has been linked to the induction of the ER unfolded protein response (UPR) stress pathway attendant with hypertension and atherosclerosis [84]. Catalase deficient mice (deficient in the turnover of peroxisomal H_2_O_2_) also show an elevated UPR [85]. What remains to be determined is whether a physical presence of H_2_O_2_ derived from non-ER sources is being directly sensed at the ER (e.g., through thiol switches), or whether the impact on the ER is a downstream, indirect outcome, relying on additional events and mediators. With the continual advent of tools to monitor H_2_O_2_ diffusion, and the ability to detect the oxidation of macromolecules by H_2_O_2_, it will be exciting to follow the potential detection of additional ER-organelle ROS nanodomains. Notably, redox-signaling events need not be limited to the ER and a single proximal organelle; for example, a proposed redox triangle between the ER, mitochondria, and peroxisome has been outlined [86].

## 4. ER Targets of H_2_O_2_

As introduced above, H_2_O_2_ production and consumption are well integrated into the fundamental cellular processes carried out by the ER. The ER is the primary site for the folding and assembly of proteins that are ultimately distributed throughout the secretory pathway, delivered to the plasma membrane, and secreted into the extracellular space [87]; H_2_O_2_ is both generated and consumed during the formation of disulfide bonds in folding nascent chains. Similarly, the ER is involved in the biogenesis of lipids and steroid hormones, which are subsequently distributed throughout the cell; CYP activities are essential for these pathways. Moreover, the ER serves as a tightly regulated intracellular calcium store [88], and regulated calcium release by the ER can be influenced by the presence of ROS. Below we focus on the developing understanding of how fluctuations in H_2_O_2_ production at the ER from these various sources are sensed. We concentrate our discussion on the outcomes for peroxide sensing at the ER, including the emerging mechanisms that act to couple H_2_O_2_ generation at the ER with the modulation and/or preservation of essential ER functions.

It is generally agreed that many signaling events initiated by H_2_O_2_ serve to alter cellular metabolism in response to ROS and/or change cellular activities to limit or alleviate the potential damage associated with excessive levels of ROS. However, the mechanisms for signal transduction remain poorly defined. At the ER, several proteins with cysteine residues susceptible to direct oxidation by H_2_O_2_ have been identified. In this section, we outline what is currently known about three ER proteins susceptible to oxidation by H_2_O_2_. We discuss what leads to the modification of these proteins by H_2_O_2_, the emerging downstream consequences of protein oxidation, and some of the outstanding questions relating to the pathways involving these enzymes.

### 4.1. The Molecular Chaperone BiP

BiP (GRP78) is the ER-localized member of the Hsp70 chaperone family [89], and is often described as a “master regulator” of ER functions [90,91,92]. The “master” moniker refers to the many roles BiP plays in ensuring ER homeostasis, including helping to bring proteins into the ER through the translocon, facilitating the turnover of terminally unfoldable proteins, and acting to modulate the UPR stress response pathway. BiP expression is essential to life; mammals lacking BiP fail to develop [93] and yeast strains lacking BiP (alternatively known in yeast as Kar2) are inviable [94,95]. BiP structure and function is well conserved throughout eukaryotes. A feature shared by the BiP orthologs is a conserved cysteine residue proximal to the site of nucleotide binding. Several mammalian BiP orthologs contain an additional conserved cysteine at the junction between the nucleotide-binding domain (NBD) and the substrate-binding domain (SBD).

The conserved cysteine in the BiP NBD is susceptible to reversible oxidation by H_2_O_2_ (Figure 4). BiP modification ensues when lumenal ER peroxide levels increase, which can occur as a byproduct of several events. Activity of the disulfide bond-forming enzyme ERO1 is regulated post-translationally [96,97,98], which helps moderate the amount of H_2_O_2_ produced by ERO1. When ERO1 activity is de-regulated in yeast cells (achieved through overexpression of a constitutively active mutant of ERO1), the increased activity of ERO1 results in the formation of a sulfenic acid at the conserved BiP NBD cysteine (yeast BiP Cys63) [99]. BiP is presumed to detect the increased H_2_O_2_ generated by the de-regulated ERO1 directly; it is generally accepted that cysteine oxidation by peroxide does not require an enzyme mediator. Oxidation of BiP in the ER lumen is also observed upon treatment of cultured yeast cells with peroxide [99]. Treatment of recombinant mammalian (mouse) BiP protein with peroxide has established that the analogous cysteine is susceptible to direct oxidation; the presence of a sulfonic (–SO_3_H adduct) was reported for purified BiP treated with an excess of H_2_O_2_ [100]. The formation of a sulfonic acid adduct, versus a sulfenic acid, likely reflects the high level of H_2_O_2_ used in the treatment of the recombinant protein in the absence of any peroxidases, driving the cysteine towards overoxidation.

A unique pathway to NBD cysteine oxidation has been outlined also for mammalian BiP, wherein BiP oxidation is a downstream event coupled to the direct oxidation of the peroxidase GPx7 (also known as NPGPx) [100]. Focusing on mouse BiP, Wei et al. [100] proposed that oxidation of GPx7 by H_2_O_2_ results in the transfer of a disulfide bond from oxidized GPx7 to BiP via a thiol-disulfide exchange reaction. Here the outcome is an intermolecular BiP disulfide connecting the conserved NBD cysteine (Cys41) with the second conserved cysteine found in mammalian orthologs (mouse Cys420) [100]. In vitro treatment of recombinant mouse BiP with peroxide also yields an intramolecular Cys41-Cys420 disulfide [100]. Whether a route to an intramolecular disulfide via direct H_2_O_2_ modification occurs in cells remains unknown. It is possible that both GPx7-dependent and GPx7-independent (direct peroxide-mediated disulfide formation) pathways for mouse BiP disulfide bond formation occur in cells.

Cysteine-glutathione adducts (–SSG) have also been identified at the NBD cysteine in both yeast and mammalian BiP. Glutathionylation can be a consequence of further modification of the sulfenic acid adduct by reduced glutathione (GSH) or direct oxidation by oxidized glutathione (GSSG) [101]. Glutathionylated mammalian BiP has been recovered in large-scale proteomic studies [102,103]. Glutathionylation of BiP, and similarly the formation of an intramolecular disulfide in mouse BiP, may limit irreversible oxidation (–SO_3_H) due to the further oxidation of BiP by peroxide.

A functional outcome for BiP oxidation is an enhanced ability to limit the aggregation of denatured proteins [99,100]. For yeast BiP, the increased chaperone activity has been shown to occur coincident with a loss of ATPase activity, and oxidized yeast BiP has been described as an ATP-independent protein holdase [99]. BiP holdase activity has been proposed to benefit cells during oxidative stress by limiting the aggregation of oxidant-damaged, mal-folded proteins [99]. In keeping with a potential cellular advantage for BiP oxidation during conditions of rising oxidants, a yeast strain containing an unmodifiable BiP allele (BiP-C63A) as the only cellular copy of BiP shows an increased sensitivity to overexpression of de-regulated ERO1 and exogenous H_2_O_2_ application [99]. Notably, the BiP-C63A mutant retains normal ATPase and chaperone activities, and the yeast strain with a BiP-C63A allele is viable and is indistinguishable from a wild-type strain when grown under non-stress conditions [99].

BiP activity is essential for many processes within the ER beyond just protein folding and assembly, and how BiP oxidation may adapt other ER activities during increased peroxide levels remains largely unexplored. The potential for additional impacts for BiP oxidation is highlighted by recent observations relating to how the interaction of BiP with the protein translocon may be altered upon oxidation. BiP has been proposed to act as a translocon “plug” that limits movement of calcium across the Sec61 protein translocon [104,105]. Recently, a new role for the Sec61 translocon as a transporter for glutathione (GSH) was outlined [106]. BiP in its reduced state was suggested to allow GSH import into the ER lumen, while a BiP allele that mimics cysteine oxidation (BiP-C63W) [99] appeared to block GSH entry [106]. A block in GSH movement into the ER (upon flooding the cytoplasm with GSH) could be correlated also with oxidation of the BiP cysteine [106]. Limiting GSH import into the ER is suggested to benefit cells exhibiting high ERO1 activity; an ability of BiP to sense peroxide generated by ERO1, and decrease GSH movement into the ER, has been proposed to limit an accumulation of oxidized glutathione in the ER, which may disrupt the redox poise and limit the ER oxidative folding capacity [106].

The reversibility, and thereby transience, of BiP oxidation is important to ensure that altered chaperone activity is not prolonged once H_2_O_2_ levels return to a basal state. Although BiP alleles that mimic constitutive oxidation benefit cells grown under overly oxidizing conditions, these same alleles put cells at a competitive disadvantage in the absence of stressor [99,101]. In yeast, the BiP nucleotide exchange factor Sil1 has been shown to reduce and remove BiP cysteine adducts [107]. The identity of a mammalian reductase remains undiscovered. A Sil1 ortholog is present in mammals [108] and the mammalian orthologs contain N-terminal cysteines [107], yet whether the mammalian N-terminal SIL1 cysteines are present after signal sequence processing and/or are active in the reduction of oxidized BiP has not been elucidated. Moreover, how Sil1 reductase activity in yeast is maintained remains unknown. What reduces the N-terminal cysteines in Sil1 to allow for reducing activity, and whether such activity is regulated to allow for an accumulation of oxidized BiP, are questions that need to be addressed.

### 4.2. The Transmembrane Protein Kinase IRE1

The type I membrane protein IRE1 is an ancient and evolutionarily conserved sensor of ER-stress. IRE1 mediates the UPR (unfolded protein response): a signaling pathway triggered when the protein folding demand exceeds the protein-processing capacity of the ER (recently reviewed in [109,110]). The IRE1 lumenal domain monitors the folding capacity within the ER. When unfolded proteins accumulate in the ER, IRE1 becomes activated and self-associates, which upregulates the kinase and RNase activities of its cytosolic domains. Activated IRE1 undergoes *trans*-autophosphorylation and mediates a splicesome-independent splicing reaction that results in the production of the transcription factor XBP1 (in mammals) or Hac1 (in yeast), and the transcription of select genes that increase the ER folding and degradation capacities [111]. Transient activation of the UPR helps cells maintain an ER protein folding capacity that matches the varying folding demands conferred by developmental and environmental events. If adaptive UPR signaling cannot establish a sufficient folding capacity, apoptosis is triggered, leading to cell death [112,113]. IRE1 is the only identified sensor for the UPR in yeast. Metazoans contain two IRE1 isoforms (IRE1α and IRE1β) and three distinct UPR sensors: IRE1, PERK, and ATF6.

A peroxide-sensing cysteine has been identified in the cytoplasmic domain of IRE1, within the kinase activation loop. IRE1 sulfenylation was first uncovered at Cys663 in the nematode *Caenorhabditis elegans* after exposure to arsenite [114] (Figure 5). Arsenite treatment was observed to increase the activity of the BLI-3 NOX, and knockdown of BLI-3 diminished arsenite-induced IRE1 sulfenylation, suggesting NOX-induced ROS as a source for the peroxide detected by IRE1 [114]. Worm IRE1 sulfenylation was observed also as a consequence of ROS production by the mitochondria in response to paraquat treatment, and upon perturbation of lumen ER enzymes, including ERO1 [114]. In mammals, an accumulation of sulfenylated IRE1α has been detected in a spontaneously hypertensive rat (SHR) model [115]. IRE1α modification was lessened upon inhibition of the ER-localized NOX4 [115]. Although it seems likely that the modified IRE1α cysteine is positionally analogous to that observed in worms, which rat cysteine is oxidized in the hypertensive rat model remains to be determined.

The impact of IRE1 sulfenylation is a dramatic re-coordination of the downstream signaling by IRE1. Formation of a sulfenic acid inhibits IRE1 kinase activity, which prevents the splicing of mRNA encoding XBP1 [114]. Alternatively, oxidation activates the activity of the worm NRF2/SKN-1 transcription factor, well established for its role in mounting an antioxidant response [114,116]. IRE1 sulfenylation generates a scaffold for recruitment of TRAF2 and the kinase NSY-1, which initiates a p38 MAP kinase cascade that results in activation of the transcription factor NRF2/SKN-1 [114]. Worms with an IRE1 mutation that cannot be sulfenylated (C663S) do not survive arsenite treatment, demonstrating an advantage for IRE1 sulfenylation [114].

A phenotypic analysis suggests the potential for an analogous oxidant response pathway in yeast, although distinctions were observed. An attenuation of canonical UPR signaling was seen in the presence of arsenite (in worms and yeast) or peroxide (in yeast), and attenuation required the conserved cytoplasmic IRE1 cysteine [114,117]. The inhibitory effect of arsenite persisted even in the presence of established chemical inducers of the UPR, consistent with an inactivation of IRE1 kinase activity [114,117]. Yet, if and/or what alternative signaling is activated in yeast upon IRE1 modification remains to be elucidated. Hog1, the homologous yeast p38 MAP kinase, is activated by arsenite exposure [117,118]; yet, activation does not require the IRE1 cysteine [117]. The ultimate transcriptional output is also bound to be different. In yeast, the transcription factor-based oxidative stress response is mediated through direct oxidation of the Yap family of transcription factors, which is distinct from the phosphorylation-based events described for SKN-1 [119]. Furthermore, no growth defect was observed for the yeast IRE1-C832S mutant in the presence of arsenite [117], leaving open the question as to what benefit may be conferred through the potential oxidation of yeast IRE1 Cys832.

Despite intriguing hints from reports focused on yeast and mammalian cells [115,117], whether sulfenylation is a conserved event used to alter the downstream signaling of IRE1 to mediate an antioxidant response remains an open question. Potential distinctions between the consequences observed for worm IRE1 sulfenylation and the potential mammalian response to IRE1 sulfenylation have been observed. The vascular cells from the SHR rat with elevated sulfenylated IRE1 also showed increased phosphorylated IRE1α and spliced XBP1 [115], distinct from the block in canonical UPR signaling observed in arsenite-treated worms [114]. Whether this reflects an inability of sulfenylated IRE1α to downregulate the canonical UPR, if the observed XBP1 splicing is a product of unmodified IRE1α, and/or if the sulfenylated IRE1α reflects oxidation at the cysteine in the kinase activation loop are unknown.

How IRE1 sulfenylation is reversed is unclear, and determining what downregulates the oxidant response will provide additional insight into this pathway. Notably, the potential for H_2_O_2_-mediated modification of IRE1 to initiate a terminal (versus adaptive) response is also intriguing. Recruitment of TRAF2 to IRE1 is best characterized in the context of pro-apoptotic signaling, where TRAF2 association with IRE1 initiates a JNK cascade that feeds into the intrinsic apoptosis pathway [120,121]. Whether adaptive and terminal signaling of oxidative stress occurs through IRE1, as reported for the canonical UPR, remains to be determined.

### 4.3. The SERCA2 Calcium Pump

The sarco/endoplasmic reticulum calcium transport ATPase (SERCA) is a P-type ATPase family member that pumps calcium from the cytosol into the ER lumen against its concentration gradient. SERCA pump activation allows cells to clear cytosolic Ca^2+^ to downregulate events like muscle contraction and restore ER Ca^2+^ for future signaling events. ER Ca^2+^ levels are intimately tied also to overall ER homeostasis. Maintaining Ca^2+^ levels in the ER is required for the activities of several ER chaperones that are governed by Ca^2+^ binding [122]. Vertebrates contain three SERCA genes: SERCA1–3. Alternative splicing generates more than 10 different protein isoforms, which show unique developmental expression and tissue distribution patterns [123,124].

Three of the SERCA2 cysteines have been identified as especially prone to oxidation: a cysteine in the cytoplasmic P domain (Cys674) and a pair of cysteines positioned in a lumenal ER loop (Cys875 and Cys887). Below we discuss first how physiological levels of H_2_O_2_ are proposed to modulate Ca^2+^ signaling through SERCA2 Cys674. Following, we discuss a second pair of lumenal SERCA2 cysteines (Cys875/887) also proposed to undergo redox-regulation.

#### 4.3.1. SERCA2 Cys674

Glutathionylated and sulfonylated forms of SERCA2 Cys674 have been detected, and oxidation has been linked to several types of ROS and ROS sources. Increased levels of glutathionylated Cys674 are reported during conditions associated with increased SERCA2 activity, including upon muscle relaxation [125,126] and during VEGF-induced endothelial cell migration [127,128]. Glutathionylation has been ascribed to the presence of peroxynitrite (ONOO^−^) derived from nitric oxide (NO) and superoxide [125]. A mechanism for glutathionylation via nitroxyl (HNO), independent of peroxynitrite, has also been suggested [126]. NOX2 and NOX4 have been implicated as sources of superoxide and/or peroxide that influence endothelial SERCA2 Cys674 glutathionylation [127]. Glutathionylation of cellular SERCA2 can be stimulated also by exogenous H_2_O_2_ addition [127].

Irreversiblity sulfonylated SERCA2 Cys674 has been recovered from cells of senescent [129], artherosclerotic [125,130], and obese [131,132] animals. SERCA2 sulfonylation is also correlated with myocardial fibrosis in human patients with non-ischemic cardiomyopathy [133]. In the Zucker diabetic rat model, sulfonylation is observed alongside elevated H_2_O_2_ and an upregulation of NOX4 [131], and NOX4 knockdown was shown to lessen the levels of H_2_O_2_ and SERCA2 sulfonylation [131]. In senescent mice, modification of SERCA2 was decreased upon overexpression of catalase, which reduces H_2_O_2_ to water [129].

Dual outcomes are described for SERCA2 Cys674 oxidation, dependent on the nature of the redox modification at Cys674 (Figure 6). Reconstitution experiments with purified SERCA2 demonstrate glutathionylation of Cys674 facilitates activation of SERCA2 and increased ER Ca^2+^ uptake [125]. Overexpression of wild-type or a C674S mutant of SERCA2 in HEK293 or human aortic endothelial cells establish Cys674 as the glutathionylation site [125] and that Cys674 modification is linked to an increase in SERCA2 activity [125,127,128,130]. Relaxation of rabbit carotid artery (in response to NO), inhibition of vascular smooth muscle migration (by NO), and increased endothelial cell migration and tube formation (in response to VEGF), can be correlated with the presence of glutathionylated SERCA2 [125,128,130]. Consistent with the importance for glutathionylation in SERCA function, mice containing a heterozygous SERCA2 C674S knock-in (SKI mice) exhibit impaired developmental and ischemic angiogenesis [134].

Conversely, SERCA2 Cys674 sulfonylation is associated with a decrease in SERCA2 activation. An increase in sulfonylated Cys674 in artherosclerotic aorta samples correlates with a decrease in glutathionylated SERCA2 and a lack of stimulated Ca^2+^ uptake upon NO addition [125]. Similarly, the increased leveled of sulfonylated SERCA2 in diabetic mouse smooth muscle correlates with abnormal vascular smooth muscle cell migration [131]. As mentioned above, overoxidation is associated with aging and several disease pathologies. It has been proposed that the transition between a SERCA2 with the capacity to undergo reversible glutathionylation, and a SERCA2 unable to respond to redox changes upon overoxidation, reflects a movement between physiological and pathological levels of ROS [125]. The decreased responsiveness of sulfonylated SERCA2 to NO may help explain the decrease in NO-dependent vascular relaxation associated with biological aging.

What reverses the glutathionylation of SERCA2 to modulate SERCA2 activity remains to be elucidated. Cysteine sulfonylation is considered an irreversible modification, and overoxidized SERCA2 may be subject to cellular turnover via degradation [135]. What facilitates (or limits) the accumulation of overoxidized SERCA2 needs also to be uncovered.

#### 4.3.2. SERCA2 Cys875 and Cys887

SERCA2 Cys875 and Cys887 are located in a lumenal loop (L4) connecting TMD 7 and 8, and this pair of cysteines has been implicated in redox regulation of SERCA2 calcium pump activity. Multiple oxidation forms for the Cys875/887 pair have been detected or inferred, including sulfenylated or disulfide-bonded states. H_2_O_2_ generated from overexpressed ERO1α has been linked to SERCA2 cysteine sulfenylation [136]. SEPN1 is an enzyme implicated in reversing sulfenylated SERCA2 cysteines [136]. A decreased ability to isolate a SERCA2–SEPN1 complexes from cells containing a Cys875/887 mutant, implies the L4 cysteines are the site of SERCA2 sulfenylation [136]. A 50% decline in a free thiol at SERCA2 Cys875 (Cys875 oxidation) was observed also in proteomic studies of mouse cardiac lysate upon treatment with H_2_O_2_, though what reversible oxidation state occurred at Cys875 was not established [137]. Overexpressed SERCA2b has been recovered with predominantly reduced L4 cysteines under basal growth conditions, and these cysteines shifted to predominantly oxidized state upon knockdown of the reductant ERdj5 [138]. Given the propensity for ERdj5 to reduce disulfide bonds, it has been proposed that ERdj5 maintains SERCA2 in a reduced state [138].

Oxidation of Cys875 and/or Cys887 has been linked to the negative regulation of SERCA2 activity. Knockdown or knockout of SEPN1 or ERdj5 (both implicated in SERCA2 Cys875/887 reduction) correlates with decreased SERCA2 activity and lower [Ca^2+^]_ER_ [136,138,139,140]. Overexpression of a mammalian SERCA2 with mutated cysteine residues 875 and 887 (reasoned to behave like a reduced SERCA2) in *Xenopus* oocytes increased Ca^2+^ oscillation frequency relative to that observed with wild-type SERCA2, suggesting an increase in SERCA2 pumping [141]. Reduction of the L4 cysteines appears to be regulated in response to calcium availability in the ER lumen. SEPN1 and ERdj5 adopt inactive oligomeric states when lumenal calcium levels are sufficiently high [138,142]. Activation of the SEPN1 and ERdj5 reductases under low [Ca^2+^]_ER_ is proposed to help refill depleted ER calcium stores. Coordinated expression of ERO1α and SENP1 has led to the speculation that SENP1 may work also to prevent dysregulation of Ca^2+^ homeostasis as a consequence of ERO1-generated peroxide and hyper-oxidation of lumenal SERCA2 cysteines [136]. Consistent with this potential role for SEPN1, overexpression of ERO1α compromises muscle function in SEPN1 KO [136,143].

Two members of the PDI family (ERp57 and TMX1) have been suggested to mediate oxidation of the L4 cysteines and facilitate decreased SERCA2 activity [141,144]. When co-expressed with SERCA2b in the *Xenopus* oocyte, ERp57 reduces the frequency of calcium oscillations (decreases SERCA2 activity), and the effect of ERp57 is abolished upon mutation of the ERp57 active sites, which are required for disulfide bond catalysis by ERp57 [141]. TMX1 overexpression correlates with decreased Ca^2+^ uptake by the ER, and the activity of TMX1 required its active site cysteines. However, it should be noted that the impact of ERp57 or TMX1 levels has not been linked directly to L4 cysteine oxidation [141,144]. Moreover, the upstream signals that may modulate oxidation of SERCA2 by ERp57 or TMX1, and whether Ca^2+^ levels and/or H_2_O_2_ regulate the interaction of these enzymes with SERCA2, remain largely unclear. A model for Ca^2+^-regulated association of ERp57 and SERCA2 has been proposed, wherein a complex of ERp57 with the Ca^2+^-binding chaperones calnexin or calreticulin modulates interaction with the L4 loop dependent on [Ca^2+^]_ER_ [141]. Interestingly, the impact of ERp57 on Ca^2+^ oscillations was observed for SERCA2b but not SERCA2a, and it was postulated that the isoform specificity may relate to the divergent C terminus in SERCA2b [141]. It will be exciting if one can confirm an isoform specific regulation via the lumenal cysteines that connects to the unique structural conformations influenced by the C terminus [145,146]. However, not all studies suggest isoform specific oxidation/reduction of SERCA2; SEPN1 was shown to associate with both SERCA2a and SERCA2b [136].

## 5. Conclusions and Future Directions

An increasing number of studies focused on the importance of H_2_O_2_ as a signaling molecule, and the ER as a source of ROS and redox-based signaling events, have appeared in the literature over the last two decades. Advances in mass spectrometry and the chemical probes for the detection of cysteine adducts suggests that 10–20% of the proteome thiols are susceptible to post-translational cysteine oxidation [24]. A recent proteomic study focusing on an enriched ER fraction identified ~900 reactive cysteines within secretory proteins; ~500 of these proteins were characterized as partially oxidized, suggestive of controlled oxidation events versus stable disulfide bonds [147]. Yet, an understanding of the mechanistic and physiological outcomes for individual protein oxidation events has emerged for only a small fraction of those proteins identified as susceptible to oxidation. In addition, connecting specific physiological sources of ROS to the oxidation of a select protein target remains a work in progress.

As outlined above, significant strides have been made in parsing these connections for several ER proteins. These studies reveal biochemical and metabolic changes initiated in response to H_2_O_2_ levels that can increase cell survival. We foresee similar unique pathways will emerge from the further characterization of other ER targets of ROS. In addition, we anticipate future research efforts will begin to address the outstanding questions related to redox modification of BiP, IRE1, and SERCA2, including some of the unknowns highlighted in the main text relating to conservation of these pathways across species, sources of H_2_O_2_ for modification, and the yet-to-be identified enzymes that facilitate reversibility. One expects the next decade will bring further insights into mechanisms for redox signaling by BiP, IRE1, and SERCA2 and also an understanding of the crosstalk between these pathways.

## Figures and Tables

**Figure 1 cells-09-02314-f001:**
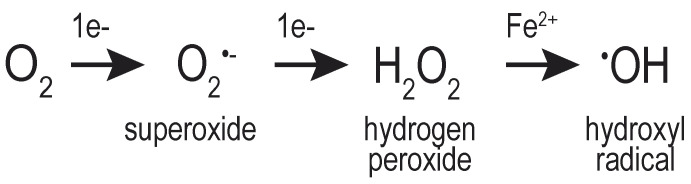
Reactive oxygen species. Scheme shows several of the types of ROS generated through the successive reduction molecular oxygen (O_2_).

**Figure 2 cells-09-02314-f002:**
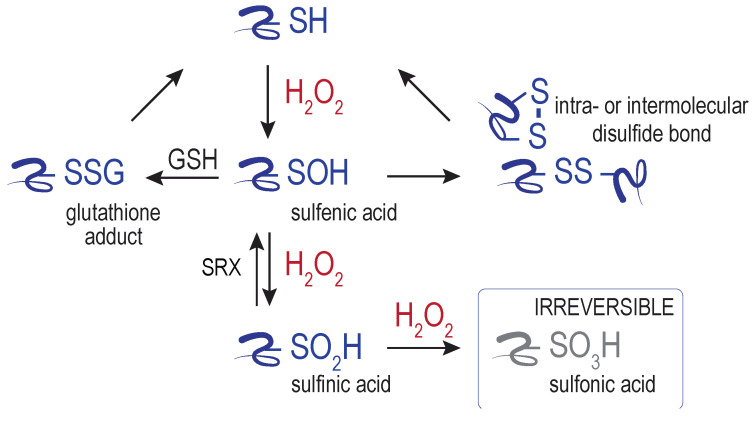
Cysteine modifications initiated by H_2_O_2_. Hydrogen peroxide (H_2_O_2_) reacts with target protein cysteine thiol (–SH), producing a sulfenic acid adduct. Subsequent reactions with glutathione (GSH), protein thiols, or H_2_O_2_ can generate additional modifications. Glutathione adducts, disulfides, and sulfinylated proteins can be reduced to the free thiol state through the action of proteins and low-molecular weight thiols [4]. The only established route for reduction of sulfinylated cysteines is by sulfiredoxin (SRX). Sulfonylated cysteines are considered irreversible oxidation products. This figure depicts the further modification of sulfenic acid by thiols (GSH, proteins) or H_2_O_2_. However, sulfenic acids can react also with a backbone amide (forming sulfenylamide), condense with another sulfenic acid (forming a thiosulfinate), or undergo further modification by hydrogen sulfide; these modifications are not shown here and are further discussed in [4,25,26].

**Figure 3 cells-09-02314-f003:**
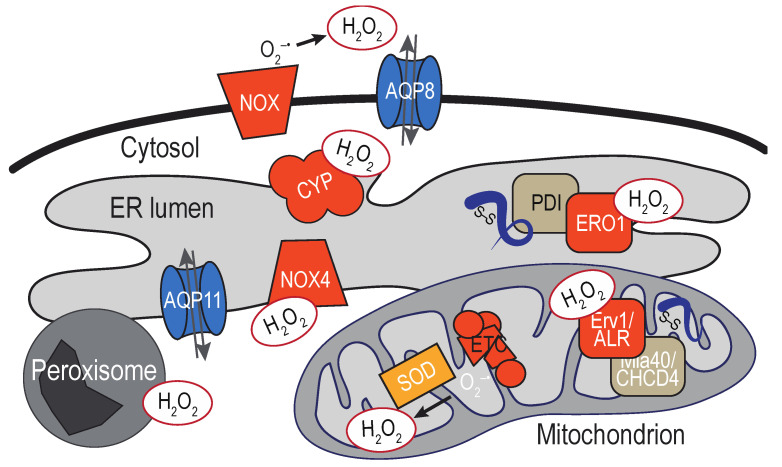
Sources of H_2_O_2_ at the endoplasmic reticulum (ER). Several direct and indirect enzymatic sources of H_2_O_2_ are depicted and colored in dark orange (refer to text for more details). Two examples of peroxiporins that facilitate bi-directional peroxide transport across membranes are noted in blue.

**Figure 4 cells-09-02314-f004:**
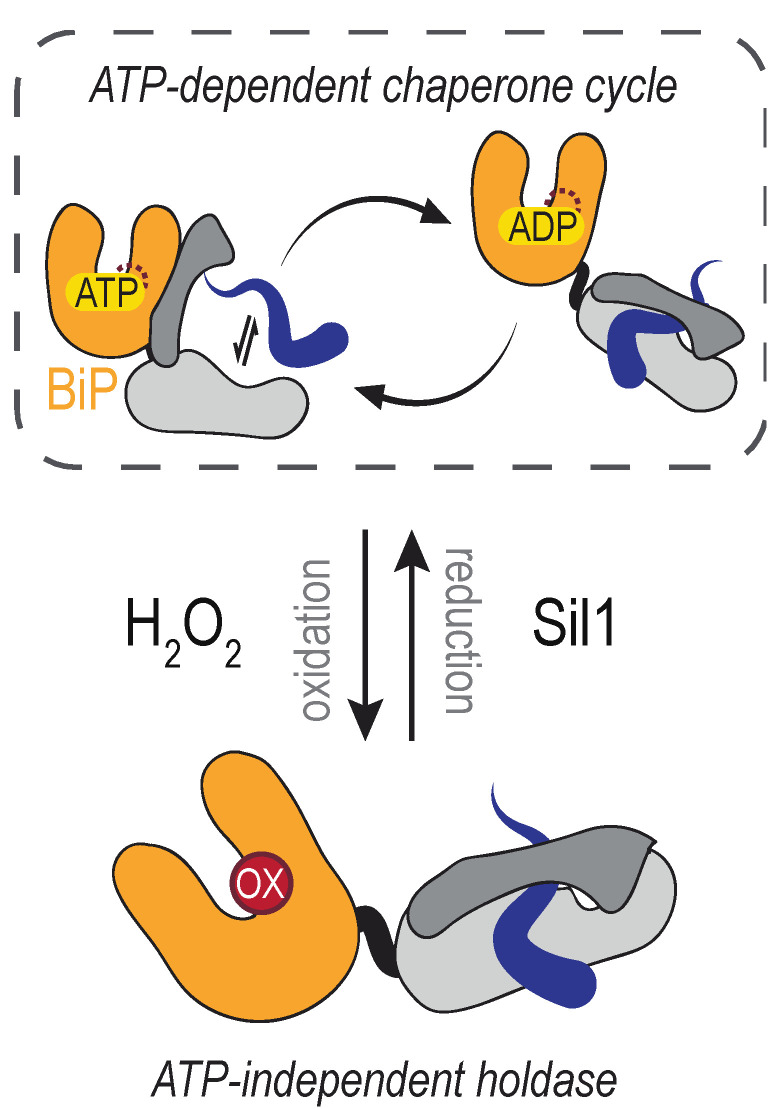
The *Saccharomyces cerevisiae* Hsp70 BiP is reversibly modified by H_2_O_2_. A conserved cysteine (red circle) in the nucleotide-binding domain (NDB, orange) of yeast BiP can be oxidized in the presence of elevated H_2_O_2_. Upon oxidation, BiP ATPase activity is inhibited yet the peptide-binding domain (grey) maintains an ability to hold polypeptides (blue). Oxidation of BiP can be reversed by Sil1, which restores the canonical Hsp70 chaperone cycle.

**Figure 5 cells-09-02314-f005:**
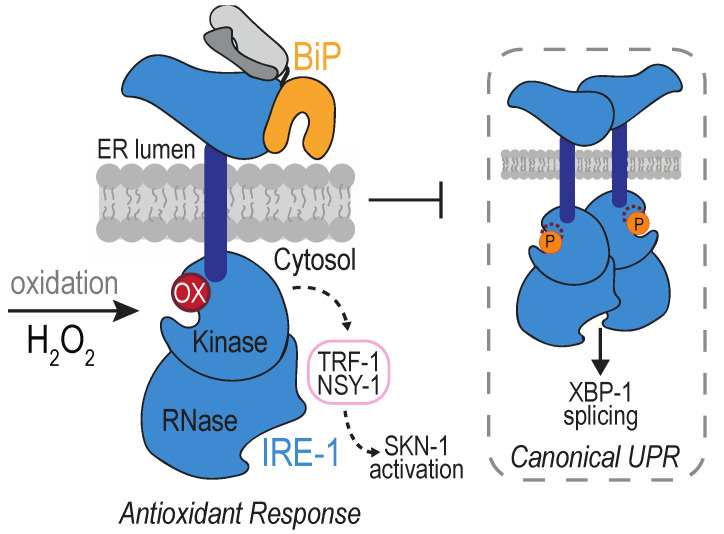
Oxidation of the *Caenorhabditis elegans* IRE-1 kinase domain inhibits canonical unfolded protein response (UPR) signaling and promotes an antioxidant response. High levels of hydrogen peroxide (H_2_O_2_) triggers oxidation of a conserved cysteine (red circle) in the activation loop in the kinase domain, which inhibits kinase activity and prevents activation of the RNase domain, preventing canonical UPR signaling. Dissociation of BiP from IRE-1 is normally associated with UPR induction, and BiP remains associated with oxidized IRE-1 [114]. Oxidation results in the recruitment of the scaffold TRF-1 and kinase NSY-1, which mediate initiation of an antioxidant transcriptional response through the transcription factor SKN-1.

**Figure 6 cells-09-02314-f006:**
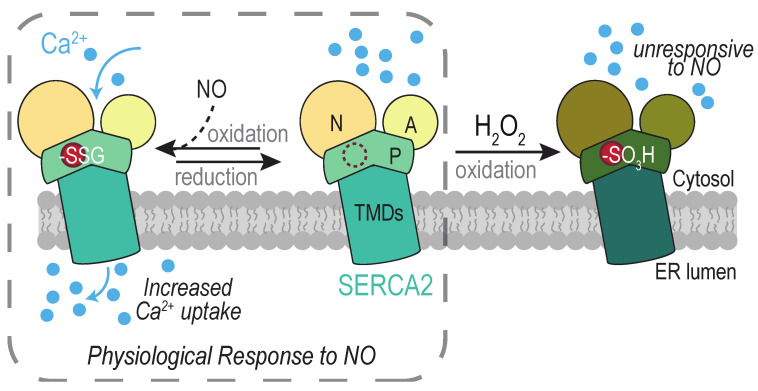
Mammalian SERCA2 oxidation at the cytosolic Cys674 modulates Ca^2+^ uptake into the ER in response to nitric oxide (NO). Cys674 (red circle) is located within the SERCA2 P domain, between the nucleotide binding domain (N) and the transmembrane domains (TMDs). The presence of NO in combination with cellular ROS and reduced glutathione results in SERCA2 glutathionylation (–SSG) and increased pumping of Ca^2+^ from the cytosol into the ER lumen. An accumulation of sulfonylated SERCA2 (–SO_3_H) is associated with a lack of responsiveness to NO and various disease pathologies, including artherosclerosis. A, actuator domain.

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
