# Peer review of "Pathways for Sensing and Responding to Hydrogen Peroxide at the Endoplasmic Reticulum"

_cells, 2020, doi:10.3390/cells9102314_

Round 1

Reviewer 1 Report

This is a beautifully written and illustrated review that covers a very important emerging topic, and it was nice to see it all pulled together.  I have only two major comments, each about Figure 2:

--In Figure 2 it is indicated (as is usually done) that sulfinylation is reversible, but as far as we know currently this occurs only within peroxiredoxins and is mediated by a sulfiredoxin.  This is an important distinction from sulfenylation, which has multiple routes to reversibility.

--An emerging area that is likely to be very important for this topic is the extent, effects, and control of persulfidation, and how H2S may thereby act as a protective agent that modifies sulfenylated cysteines to -SSH.  This readily modified species may represent the most important intermediate for reversibility of -SOH and prevention of further oxidation.  See PMID: 31735592

 for a recent story along these lines that is very exciting. 

--line 398, is SNK1 supposed to be SKN-1?

Reviewer 2 Report

Well written and structured review about an exciting field of on-going research work. The specific topics covered by the review are well chosen and contain the major player in the field of ER-related pathways for sensing and responding to hydrogen peroxide. The authors included findings from different types of models including C. elegans, yeast and mammalian cells, which enables the reader to have an excellent overview about the whole spectrum of redox-switches in the mentioned enzyme orthologs.

Minor points:

p. 4, l. 136-144: In addition to the cited papers the authors should mention the new approach published by Melo et al. using a HyPer variant for the detection of H2O2 inside the ER:

TriPer, an optical probe tuned to the endoplasmic reticulum tracks changes in luminal H2O2, Eduardo Pinho Melo, Carlos Lopes, Peter Gollwitzer, Stephan Lortz, Sigurd Lenzen, Ilir Mehmeti, Clemens F. Kaminski, David Ron, Edward Avezov, BMC Biol. 2017; 15: 24. Published online 2017 Mar 27. doi: 10.1186/s12915-017-0367-5, PMCID: PMC5368998

p.5, l. 186: The statement that “the mitochondria contain several peroxidases that convert H2O2 to water” is not covered by the cited literature (61: Finger, Y., Riemer, J. Protein import by the mitochondrial disulfide relay in higher eukaryotes. Biol Chem 675 2020, 401, 749-763). Did the authors confuse the cited literature with another publication?

Reviewer 3 Report

The review article entitled "Pathways for sensing and responding to hydrogen peroxide at the endoplasmic reticulum" by Roscoe and Sevier is comprehensively described recent studies about importance of hydrogen peroxide at the endoplasmic reticulum (ER). It is summarized the current view of hydrogen peroxide in ER and three targets of reversible post-translational modification by ER peroxide: BiP (Grp78), IRE1, and SERCA2. The manuscript is well-written and figures are informative.

Very recent comprehensive review that is similar to this manuscript on reactive oxygen species including hydrogen peroxide by Sies and Jones (2020) focused on the role of ROS as physiological signaling agents (Ref #3). The authors contribute to a different aspect of this field. They described how hydrogen peroxide in endoplasmic reticulum affects the function of ER chaperone Bip and ER stress sensor IRE1. This paper also summarize recent findings demonstrating regulation of the SERCA2 calcium pump by oxidation at physiological and pathological levels. 

This review is a well-written paper on a topic of interest to cell biologists. Only minor points need to be addressed:

The authors may include the following published paper.

  1. Reczek CR and Chandel NS, ROS-dependent signal transduction, Current Opinion in Cell Biology, 33, 8-13 (2015)
  2. Kakihara T, Nagata K, Sitia R, Peroxides and peroxidases in the endoplasmic regiculum: Integrating redox homeostasis and oxidative folding, Antioxidants and Redox Signaling, 16, 763-771 (2012)
  3. Finkel T, From sulfenylation to sulfhydration: What a thiolate needs to tolerate, Science Signaling, 5, 215 (2012)
